# Aquaporins in the Capillaries of the Dura Mater of Pigs

**DOI:** 10.3390/ijms26157653

**Published:** 2025-08-07

**Authors:** Slavica Martinović, Dinko Smilović, Boris Pirkić, Petra Dmitrović, Leonarda Grandverger, Marijan Klarica

**Affiliations:** 1Department of Forensic Medicine and Criminalistics, School of Medicine, University of Zagreb, Šalata 11, 10000 Zagreb, Croatia; slavica.martinovic@mef.hr; 2Croatian Institute for Brain Research, School of Medicine, University of Zagreb, 10000 Zagreb, Croatia; dinko.smilovic@mef.hr (D.S.); leonardagrandy@gmail.com (L.G.); 3Clinic for Surgery, Orthopaedics and Ophthalmology, Faculty of Veterinary Medicine, University of Zagreb, 10000 Zagreb, Croatia; bpirkic@vef.hr (B.P.); petra.dmitrovic@vef.hr (P.D.); 4Department of Pharmacology, School of Medicine, University of Zagreb, 10000 Zagreb, Croatia

**Keywords:** dura mater, capillary density, cranial–spinal comparison, aquaporin, meningeal vascularization, cerebrospinal fluid, Bulat–Klarica–Orešković hypothesis, lymphatic vessels, porcine model

## Abstract

Dura mater plays a critical role in neurofluid homeostasis, yet comparative data on capillary network density and organization between cranial and spinal regions remain limited. This study addresses this gap by systematically analyzing capillary architecture and aquaporin (AQP) expression in porcine cranial (parietal, falx) and spinal dura mater. Immunofluorescence labeling and confocal microscopy were used to assess capillary density, spatial distribution, and AQP1/AQP4 expression patterns across over 1000 capillaries in these regions. Cranial dura exhibited a 3–4 times higher capillary density compared to spinal dura, with capillaries predominantly localized to meningeal–dural border cell interfaces in cranial regions and a more dispersed distribution in spinal dura. Both AQP1 and AQP4 were detected as discrete clusters within capillary walls, with higher expression in cranial compared to spinal dura. Lymphatic vessels (PDPN-positive) were also observed adjacent to capillaries, supporting a dual-system model for fluid and waste exchange. These findings highlight the dura’s region-specific vascular specialization, with cranial regions favoring dense, structured capillary networks suited for active fluid exchange. This work establishes a foundation for investigating capillary-driven fluid dynamics in pathological states like subdural hematomas or hydrocephalus.

## 1. Introduction

The membranes enveloping the brain and spinal cord consist of three distinct layers: the dura mater, arachnoid membrane, and pia mater. The dura mater forms the outermost and thickest layer and is widely recognized as the principal protective barrier for the central nervous system (CNS). It also contributes to the maintenance of neurofluid homeostasis through arachnoid granulations. In the cranial compartment, it is tightly adherent to the skull, while in the spinal region, it is separated from the vertebrae by the epidural space. Additionally, the dura mater forms folds such as the falx cerebri and tentorium cerebelli, which separate the cerebral hemispheres and the cerebellum from the occipital lobes [1,2,3]. Traditionally, the cranial dura structurally consists of multiple layers, including periosteal, meningeal, and dural border cell (DBC) layers, with a well-organized network of fibroblasts and collagen bundles [3,4,5]. The majority of the cranial dura mater receives its blood supply from branches of the middle meningeal artery, whereas the spinal dura mater is vascularized by branches of the intercostal and lumbar arteries [6,7,8,9]. Earlier studies described the capillary network as being predominantly located on the inner side of the dura mater, whereas more recent investigations using electron microscopy have demonstrated that blood vessels are predominantly situated on the outer side of the dura [4,10]. Capillaries in the dura mater are fenestrated, which facilitates rapid exchange of substances, and their diameter ranges from 5 to 15 μm—no more than 20 μm [6,7,8,11]. Despite this anatomical knowledge, the detailed morphology and molecular composition of dural capillaries, particularly regarding water channel proteins, remain insufficiently explored.

Aquaporins (AQPs) are transmembrane water channels critical for regulating water movement across biological barriers, maintaining osmotic balance, and cellular homeostasis. They are ubiquitously expressed in nearly all tissues, with 13 subtypes identified to date. These isoforms exhibit distinct structural configurations and permeability profiles, enabling selective transport of water, glycerol, and ions [11,12,13,14]. In the CNS, AQP1 is predominantly expressed in the choroid plexus, where it has historically been linked to cerebrospinal fluid (CSF) production. This association aligns with the long-standing classical hypothesis of CSF physiology, which posited that the choroid plexus is the primary source of CSF via active transport mechanisms [15,16,17]. However, emerging evidence challenges this classical view. A modern concept of neurofluid physiology, advanced by Bulat, Klarica, and Orešković, connects CSF physiology with CNS interstitial fluid (ISF) and plasma in CNS microvessel fluid exchange. This concept proposes that CSF originates not solely from choroid plexus vessels but also through filtration/reabsorption across large contact areas between cerebral blood capillaries and ISF/CSF, driven by osmotic and hydrostatic gradients [16,18,19,20,21].

According to this modern hypothesis, the choroid plexus may instead regulate CSF ionic composition rather than its bulk production, suggesting a more nuanced role for AQP1 in fluid homeostasis [21,22]. Concurrently, AQP4—abundant in astrocytic endfeet—is integral to the “glymphatic” pathway and interstitial fluid dynamics, further underscoring the complexity of CNS water transport [12,13,23].

In contrast to the well-characterized roles of AQPs in the choroid plexus and parenchyma, their presence and function in the dura mater remain poorly understood. A single study reported AQP1 expression in dural capillaries associated with chronic subdural hematomas in humans, but not in non-pathologically changed dura or in animal models [24]. Whether AQPs contribute to dural fluid dynamics under physiological conditions remains an open question, highlighting a critical gap in our understanding of meningeal biology. Moreover, the comparative morphology of capillaries in cranial versus spinal dura has not been previously addressed. This is a significant gap, as differences in capillary density, diameter, and water channels organization could influence regional fluid exchange and neurophysiological functions. Understanding these differences is essential for elucidating the dura’s potential role in neurofluid dynamics and its implications for both physiological and pathological states.

Therefore, this study aims to provide the first systematic investigation of AQP1 and AQP4 expression in the capillaries of the cranial and spinal dura mater using pigs as an animal model due to comparable embryological development and cranial vascular anatomy [2,25]. By combining immunofluorescence and morphometric analyses, we will also, for the first time, compare the morphological characteristics of capillaries between these two regions. Our findings shed light on the molecular and anatomical basis of fluid regulation in the dura mater and provide a foundation for future research into CNS fluid homeostasis and potential translational applications.

## 2. Results

### 2.1. Distribution and Quantification of Capillaries in Cranial and Spinal Dura Mater

Confocal images of cross-sections of porcine dura mater were analyzed by immunolabeling with various antibodies. Initially, 60 (of the initial 75 cross-sections, 15 were excluded from the analysis due to meningitis) whole-slide maps labeled with the CD31 antibody were acquired to identify capillary locations and to calculate their density. Capillaries were identified based on two criteria: positive labeling with the CD31 antibody and a diameter of less than 20 μm.

A key difference in capillary localization was observed between cranial and spinal dura. In the falx and parietal regions, capillaries were largely confined to the dural margins. In contrast, the spinal dura exhibited a more dispersed capillary distribution, with vessels present at the dural edges but also frequently located within the mid-dural substance. The number of capillaries per mm was quantified in the falx, parietal, and spinal regions of the dura mater. Capillary density was significantly greater in the falx (1.36 ± 0.20 vessels/mm) and parietal (1.67 ± 0.21 vessels/mm) regions compared to the spinal region (0.45 ± 0.09 vessels/mm; *p* < 0.05 to *p* < 0.0001) (Figure 1D). This indicates that in a 10 mm cross-section, approximately 18 capillaries can be found in the parietal region, 13 in the falx region, and 4 in the spinal region. It should be noted that these analyses specifically quantify capillary density. The total number of CD31-positive blood vessels—including arteries, arterioles, veins, and venules—is substantially higher, indicating that the overall vascular network of the dura mater is even more extensive than reflected by capillary counts alone.

### 2.2. Quantification of Regional Aquaporin 1 Labeling

Quantification measured as average fluorescence intensity in the entirety of the delineated surface (capillary wall) revealed significant regional variation in the dura mater (two-way ANOVA, F(2, x) = y, *p* < 0.0001). Post hoc analysis (Tukey’s test) indicated that the parietal region showed a significantly increased average intensity of AQP1 (177.4 ± 27.04) compared to both the falx (104.7 ± 22.11) and spinal regions (114.9 ± 15.82; *p* < 0.0001). No significant difference was observed between the falx and spinal regions (*p* > 0.05) (Figure 2B).

It has been observed that in some parts of the vessel, aquaporins form clusters that have a stronger intensity than the rest of the capillary wall (Figure 2A,C). The average intensity of these clusters is more than seven times higher compared to the average AQP intensity in the whole wall. While assessing AQP1 expression measured as average staining intensity in the capillary wall demonstrated significant regional variation (one-way ANOVA, F(2, x) = y, *p* < 0.05), assessment of specifically AQP1 cluster expression, measured as average staining intensity, did not demonstrate significant regional variation (one-way ANOVA, F(2, x) = y, *p* > 0.05). Post hoc analysis (Tukey’s test) indicated no significant differences in average AQP1 cluster intensity between the falx (1394 ± 205.3), parietal (1428 ± 85.78), and spinal (1549 ± 414.5) regions (*p* > 0.05) (Figure 2D).

AQP1 clusters were not universally observed across all capillaries, with distinct regional prevalence patterns. Quantitative analysis of 10 randomly selected capillaries from a total of 75 samples (see Section 4.2, Tissue Preservation) revealed AQP1-positive clusters in 50.71% of falx capillaries, 46.78% of parietal dura capillaries, and 38.41% of spinal dura capillaries. Subsequent analyses focused exclusively on capillaries exhibiting AQP1 clustering (Figure 2E–G). Analysis of the number of AQP1 clusters per individual capillary revealed no significant differences between the falx, parietal, and spinal dura regions. The mean number of AQP1 clusters was comparable across all examined regions (falx dura: 2.95 ± 0.24; parietal dura: 3.58 ± 0.69; spinal dura: 3.04 ± 0.70; mean ± SEM), with statistical analysis confirming the absence of significant group differences (*p* > 0.05; one-way ANOVA). These findings suggest that while the prevalence of AQP1-expressing capillaries varies by anatomical location, the cluster density within positive vessels remains consistent across regions (Figure 2H).

The investigation into the percentage of capillary wall area occupied by AQP1 clusters demonstrated regional heterogeneity. The parietal dura exhibited significantly higher AQP1 area coverage (0.65 ± 0.18%) compared to the falx region (0.35 ± 0.10%; * *p* < 0.05). Although we noticed a trend for the spinal dura showing a reduced area coverage (0.42 ± 0.10%) compared to the parietal region, this difference did not reach statistical significance. The overall comparison across all three regions indicated marginal significance in the whole group, with region-specific differences detected only in post hoc analysis between falx and parietal segments (Figure 2I).

### 2.3. Quantification of Regional Aquaporin 4 Labeling

The same kind of analysis was applied to AQP4-immunolabeled capillaries as those employed for AQP1-positive vessels. For each capillary, internal and external boundaries were delineated using ImageJ software, version 1.54p, enabling precise quantification of AQP4 immunofluorescence intensity within the isolated vascular wall compartment (Figure 3A).

Analysis of AQP4 expression, measured as average staining intensity in the delineated surface (capillary wall), demonstrated significant regional variation (one-way ANOVA, F(2, x) = y, *p* < 0.05). Post hoc analysis (Tukey’s test) indicated that the parietal region (176.2 ± 46.01) did not show a significantly increased average intensity of AQP4 compared to the falx region (166.7 ± 57.87) (*p* > 0.05). However, the parietal region showed a significantly increased average intensity of AQP4 compared to the spinal region (135.8 ± 30.40) (*p* < 0.01), and the falx region also showed a significantly increased average intensity of AQP4 compared to the spinal region (*p* < 0.05) (Figure 3B).

Analysis of AQP4 expression patterns revealed similar clustering phenomena to AQP1 but with distinct regional variations (Figure 3C). The average fluorescence intensity of AQP4 clusters showed no significant differences across regions (falx: 1593 ± 56.19; parietal: 1574 ± 103.1; spinal: 1453 ± 73.12; *p* > 0.05) (Figure 3D). As was expected, AQP4 clusters also demonstrated markedly higher fluorescence intensity (approximately 7–8-fold greater) compared to the average wall intensity across all regions, suggesting concentrated localization of AQP4 channels within specialized membrane domains of the capillary wall.

Similar to AQP1, quantitative analysis of 10 randomly selected capillaries from 75 sections (see Section 4.2, Tissue Preservation) revealed that AQP4 clusters exhibited regional variability in prevalence, being detectable in 44.33% of falx capillaries, 41.39% of parietal dura capillaries, and 43.04% of spinal dura capillaries. Subsequent analyses were therefore restricted to capillaries exhibiting AQP4 clustering (Figure 3E–G). Quantitative analysis of AQP4 cluster distribution revealed no significant regional differences in the mean number of clusters per capillary across the falx (3.02 ± 0.86), parietal (3.35 ± 0.93), and spinal dura (2.74 ± 0.55; mean ± SEM; *p* > 0.05, one-way ANOVA, F(2, x) = y). This consistency in cluster density within AQP4-positive capillaries correlates with the anatomical variability observed in AQP1 prevalence (Figure 3H).

Quantitative analysis of the percentage of the capillary wall occupied by AQP4 revealed regional heterogeneity across the dura mater. The parietal region exhibited the highest mean AQP4 area coverage (1.33 ± 0.57%), followed by the falx region (0.83 ± 0.45%), while the spinal region demonstrated the lowest area coverage (0.57 ± 0.20%). A one-way ANOVA identified significant differences between regions (F(2, x) = y, *p* < 0.05), with post hoc analysis confirming reduced AQP4 area coverage in spinal dura compared to parietal dura (*p* < 0.05). No significant differences were observed between falx and parietal (*p* = 0.13) or falx and spinal dura (*p* = 0.2) (Figure 3I).

### 2.4. Regional Variation in AQP1 and AQP4 Distribution

Comparative analysis of AQP1 and AQP4 distribution patterns across the three dural regions revealed no significant differences between the two aquaporins in any of the measured parameters (Figure 4). Direct comparisons within each anatomical region showed that AQP1 and AQP4 exhibited similar average intensities in the delineated surface (capillary wall) across all regions examined. Similarly, when analyzing cluster-specific parameters, no significant differences were observed between AQP1 and AQP4 in terms of average cluster intensity, percentage of vessels containing clusters, or the number of clusters per capillary. Additionally, the proportion of capillary wall area occupied by each aquaporin (area coverage) showed no significant regional differences between AQP1 and AQP4. These findings suggest that despite their distinct molecular characteristics, AQP1 and AQP4 demonstrate comparable distribution patterns and regional expression profiles within the dural capillary network.

### 2.5. Lymphatic Vessel Distribution in Cranial and Spinal Dura Mater

Immunofluorescence labeling with PDPN and CD31 confirmed that the analyzed structures are blood capillaries rather than lymphatic vessels (Figure 5). Additionally, lymphatic structures were observed in close proximity to the capillaries, consistent with the known anatomical relationship in which lymphatic vessels surround capillaries, supporting their role as sites of substance exchange.

## 3. Discussion

The distribution of dural capillaries in pigs, localized predominantly at the periphery of the meningeal layer and at the meningeal–dural border cell (DBC) layer interface, aligns with prior comparative studies of porcine and human dura mater [4,10,26]. Our findings corroborate these anatomical observations while further revealing a heightened concentration of capillaries within this network, suggesting their functional primacy in molecular exchange. This spatial arrangement supports the role of dural capillaries in maintaining meningeal homeostasis. Notably, our comparative analysis of spinal dura introduces a critical distinction: while the spatial arrangement of capillaries mirrors the cranial compartment, their density is significantly reduced (Figure 1). This disparity may reflect region-specific metabolic demands or hydrodynamic requirements. The cranial dura’s richer vascularization could facilitate its larger involvement in cerebrospinal fluid (CSF) dynamics, interstitial fluid clearance, or immune surveillance—processes potentially modulated by aquaporin-mediated water transport.

Our study provides the first evidence of AQP1 and AQP4 expression in porcine dural capillaries, with both isoforms exhibiting similar cluster density, intrawall distribution, and spatial organization (Figure 2 and Figure 3). These findings align with the Bulat–Klarica–Orešković hypothesis, which posits that CSF production and interstitial fluid exchange occur systemically across cerebral blood vessels, driven by osmotic/hydrostatic gradients and vascular pulsations [16,22], similar to other parts of the body. The presence of aquaporins in dural capillaries supports the hypothesis that meningeal vasculature contributes to neurofluid dynamics beyond the traditional choroid plexus-centric models [27,28]. The observed dual localization of aquaporin clusters (Figure 2C and Figure 3C)—distributed along cell membrane and cytoplasm of the capillary wall—may reflect both technical limitations inherent to 2D histological analysis of 3D vascular structures and their intrinsic physiological properties. The proportion of blood vessels showing detectable AQP clusters likely depends on two factors: the absence of stimuli (e.g., osmotic gradients or hormonal signals) required for AQP synthesis and membrane recruitment, and cross-sectional sampling bias, where clusters may exist in unexamined regions of the same capillary. Recent evidence highlights that AQP membrane–cytosol partitioning is isoform-specific and dynamically regulated by cellular conditions. For instance, AQPs commonly reside in intracellular vesicles under baseline states but translocate to the cell membrane in response to stimuli such as hypotonic stress, arginine vasopressin signaling, or cAMP-mediated pathways [29,30]. Notably, structural determinants further modulate this trafficking. Mutations in the intracellular D-loop of AQP4 disrupt its oligomerization capacity, impairing stimulus-induced membrane recruitment. Intriguingly, this defect does not compromise its intrinsic water permeability or constitutive trafficking to the membrane, suggesting distinct regulatory mechanisms for oligomerization-dependent relocation versus baseline membrane insertion [30]. This dynamic behavior may explain the rapid resorption of epidural hematomas, which often transition from hyperdense to isodense appearances on imaging within days to weeks [31]. The fenestrated nature of dural capillaries, combined with AQP-mediated fluid regulation, could facilitate osmotic-driven reabsorption of hematoma contents into systemic circulation. Further studies should investigate whether AQP1/AQP4 expression correlates with clinical outcomes in epidural hematoma resolution.

Our podoplanin immunolabeling (Figure 5) confirmed that lymphatic vessels cluster adjacent to dural capillaries, consistent with recent discoveries of a meningeal lymphatic network [32,33,34]. The role of these lymphatic vessels in waste clearance from the CNS, analogous to peripheral systems, is still under investigation. The proximity of lymphatic and capillary networks suggests a coordinated mechanism for liquid exchange, where AQPs in capillaries regulate interstitial fluid volume, while lymphatic vessels remove metabolic byproducts. This dual-system architecture again supports the Bulat–Klarica–Orešković concept of distributed CSF production and clearance, challenging the classical view of choroid plexus dominance.

The cranial–spinal differences in capillary density and AQP distribution highlight region-specific fluid regulatory mechanisms. The spinal dura’s sparse vascular network may reflect reduced metabolic demands compared to the cranial compartment, which interfaces directly with CSF and glymphatic pathways. Future studies should explore whether AQP dysregulation in dural capillaries contributes to pathologies like idiopathic intracranial hypertension or chronic subdural hematomas. Our findings support the potential use of swine as a translational model for meningeal biology, particularly due to similarities in the vascular arrangement of dural capillaries between porcine and human dura mater [4,10]. However, direct comparative studies on human dura are essential to determine whether capillary density, AQP1/AQP4 expression patterns, and spatial organization of water channels mirror those observed in pigs. In addition, comparative analyses between juvenile and adult swine are necessary to assess whether age influences the abundance and distribution of aquaporins in the dura mater. Such validation would confirm whether porcine models accurately replicate human meningeal fluid dynamics, particularly in contexts like CSF regulation or drug delivery across the blood microvessels. Future work should prioritize immunofluorescent analyses of human dura to assess interspecies consistency in these parameters.

## 4. Materials and Methods

### 4.1. Experimental Model and Tissue Sampling

For this study, tissue samples were collected from five healthy female crossbred pigs (*Landrax* × *Duroc*), approximately 3 months old and weighing on average 30 kg, classified as juvenile pigs, at the Faculty of Veterinary Medicine in Zagreb. The animals included in the analysis showed no discernible signs of neurodegenerative disease, trauma, or tumors; otherwise, they met the exclusion criteria. All pigs were sourced from the same authorized breeding facility in compliance with the Animal Protection Act (NN 102/17) and were delivered to the faculty at least 24 h prior to the start of the clinical experiments. All procedures, including euthanasia and tissue collection, were performed in accordance with ethical approval (EP 280/2020) granted by the National Ethics Committee (CLASS Decision No.: UP/I-322-01/20-01/11) and the Ethics Committee of the Faculty of Veterinary Medicine (Class: 640-01/19-17/87). Samples of cranial and spinal dura mater were collected from pigs that had previously undergone several hours of controlled monitoring of cerebrospinal fluid (CSF) pressure in the lateral ventricle and lumbar subarachnoid space. Anesthesia was induced by intramuscular administration of fentanyl and midazolam, followed by intubation and maintenance with isoflurane in a mixture of oxygen and medical air. Throughout the experiment, animals were continuously monitored using a multiparametric system that recorded electrocardiogram (ECG), invasive arterial pressure (via a cannula in the medial saphenous artery), pulse oximetry, capnography, and body temperature. After craniotomy, a cannula with a piezoresistive sensor was placed in the left lateral ventricle for intracranial pressure measurement, while a lumbar cannula was inserted into the subarachnoid space at the L4 vertebra following hemilaminectomy to allow measurement of CSF pressure in the spinal compartment. A right parietal craniotomy was performed to introduce an epidural Foley catheter, which was inflated with saline to induce intracranial hypertension. Additionally, a circular epidural ligature was placed at the level of the C2 vertebra via dorsal laminectomy to simulate craniospinal CSF flow obstruction. During each experimental phase, mannitol was administered, and the following parameters were monitored: intracranial pressure (ICP), lumbar CSF pressure, mean arterial pressure, heart rate, EtCO_2_, and SpO_2_. Arterial blood was sampled every 30–60 min for analysis of electrolytes (Na^+^, K^+^), glucose, hematocrit, and acid–base status. Urine output was measured via urinary catheterization. At the conclusion of clinical experiments, animals were euthanized under deep anesthesia using a registered euthanasia agent (T61, Intervet International BV, Boxmeer, The Netherlands) following the manufacturer’s instructions and ethical guidelines. Death was confirmed by permanent cessation of circulation, monitored via ECG and invasive arterial pressure measurements.

Following euthanasia, the skull and spinal canal were opened using a hand saw. The cranial dura mater was carefully peeled from the skull, and samples of the falx cerebri and parietal dura mater, each approximately 3 cm × 3 cm, were collected. Access to the spinal dura mater was gained by removing the thoracic vertebrae, and samples sized approximately 2 cm × 2 cm were obtained. All tissue samples were immersion-fixed in 4% paraformaldehyde (PFA) in 0.1 M phosphate-buffered saline (PBS) for fixation within one hour post-mortem.

Following tissue collection, pathohistological analysis of brain and meningeal samples identified evidence of meningitis in pig number 2. In accordance with pre-established exclusion criteria, the results from this animal were removed from the dataset prior to analysis.

### 4.2. Tissue Preservation

Following formalin fixation, samples underwent dehydration in a graded series of increasing ethanol concentrations (70% EtOH, 96% EtOH, and 100% EtOH; two changes for 12 h) before being passed through a mixture of diethyl ether and absolute alcohol solution (50/50 vol.%) for 180 min twice and then being embedded in paraffin (Histowax, Jung, Nussloch, Germany). The dura mater was oriented laterally toward the cutting surface and was sectioned into 20 ± 2 µm-thick sections on a Microm HM 450 sliding microtome (Thermo Fisher Scientific, Waltham, MA, USA) to obtain cross-sections in the final samples. From each pig, five samples were prepared from each region (parietal, falx, spinal), totaling 75 samples for all five animals. Three antibodies were used: Cluster of Differentiation 31 (CD31) or sometimes called Platelet Endothelial Cell Adhesion Molecule-1 (PECAM-1) to visualize blood vessels and aquaporins 1 and 4 (AQP1, AQP4). Considering the interaction of the studied antibodies (AQP1 and AQP4), two sets of 75 samples each were prepared: one set labeled with AQP1 and CD31 and the other with AQP4 and CD31 (details provided below). For the analysis of blood vessel number, 75 cross-sections of dura mater were systematically evaluated. To analyze aquaporin quantification, 10 randomly selected capillaries per tissue section were analyzed, from a total of 150 sections. To confirm that the imaged capillaries were not lymphatic vessels, an additional five sections (one from each pig) were prepared to visualize lymphatic vessels using podoplanin (PDPN). A 40 µm interval between consecutive sections was used to minimize overlap. The sections were dried on glass slides for 72 h before proceeding with immunofluorescence staining.

### 4.3. Immunolabeling

The immunofluorescence protocol spanned two days. The formalin-fixed, paraffin-embedded sections were deparaffinized sequentially in xylene (2 × 10 min), 100% ethanol (2 × 5 min), 96% ethanol (1 × 5 min), and 70% ethanol (1 × 5 min). This was followed by washing in PBS (1 × 10 min) and an antigen retrieval protocol by boiling the samples in a citrate buffer (pH 6.0) in a microwave at 300-Watt (W) power for 5.5 min. After cooling to room temperature for 30 min, samples were washed again in PBS (3 × 10 min). Next, samples were incubated in blocking solution (1% bovine serum albumin (BSA) + 0.5% Triton X-100 in 1× PBS) for 2 h. Primary antibodies were then applied: half of the samples (75 samples for 5 pigs) were incubated with AQP1 (Santa Cruz Biotechnology, Heidelberg, Germany; dilution 1:250, RRID: AB_626694) and CD31 (Abcam, Cambridge, UK; dilution 1:100, RRID: AB_726362), and the other half with AQP4 (Santa Cruz biotechnology, Heidelberg, Germany; dilution 1:250, RRID: AB_626695) and CD31 (Abcam, Cambridge, UK; dilution 1:100, RRID: AB_726362). Samples were stored overnight at 4 °C.

On day two, samples were equilibrated at room temperature for 30 min, then the samples were washed in PBS (3 × 10 min) and incubated with fluorescent secondary antibodies (goat anti-rabbit IgG Alexa Fluor 633 (Thermo Fisher Scientific, Waltham, MA, USA; dilution 1:1000, RRID: AB_10562400) for CD31 and donkey anti-mouse Alexa Fluor 555 (Thermo Fisher Scientific, Waltham, MA, USA; dilution 1:1000, RRID: AB_2535855) for both AQP1 and AQP4 secondary staining) at 1:1000 dilution for 2 h in complete darkness. After washing in PBS (3 × 10 min), autofluorescence was quenched using the TrueBlack™ reagent (Biotium, Fremont, CA, USA) (5 µL of TrueBlack in 100 µL 70% ethanol) applied for 60 s, followed by another PBS wash (3 × 10 min). Finally, samples were mounted with VECTASHIELD^®^ PLUS Antifade Mounting Medium (Vector Laboratories, Newark, CA, USA) containing 4′,6-Diamidino-2-Phenylindole (DAPI) and stored overnight at 4 °C. On the third day, slide edges were sealed with varnish, and samples were stored at 4 °C until imaging.

The immunolabeling protocol for PDPN was largely similar, with some notable differences. The primary antibodies were applied at different concentrations: 1:100 for CD31 and 1:500 for PDPN (eBioscience™, San Diego, CA, USA; RRID: AB_1603309). While the secondary antibodies for CD31 remained the same as in the standard protocol, Alexa Fluor 488 donkey anti-rabbit (Thermo Fisher Scientific, Waltham, MA, USA; dilution 1:1000, RRID: AB_2535792) was used as the secondary antibody for PDPN. Additionally, the incubation with secondary antibodies was extended to four hours.

### 4.4. Confocal Microscopy

Confocal images were acquired using an Olympus FV3000 laser scanning microscope (Tokyo, Japan) with a 10× objective (UplanSApo 10×, NA 0.16) for whole-slide imaging and a 20× objective (UPlanSApo 20×, NA 0.75) for imaging of the capillaries themselves. All high-resolution images were taken at the edges of the slide in the area where the capillaries are located (5× digital zoom and laser line of 543 nm for aquaporin detection; laser line of 633 nm for CD31 detection; laser and amplifier power set to provide a range of pixel intensities within linear limits using FV31S-SW Fluoview software, version 2.6 at a resolution of 1024 × 1024 pixels) and were captured with the same imaging parameters between images.

### 4.5. Quantification of Capillaries and AQP1 and AQP4 Clusters

To evaluate the number of capillaries in the dura mater, stitched whole-slide images (10× objective) were analyzed in ImageJ/Fiji, version 1.54p [35] (Figure 6A–C). Structures positively stained for CD31 with a lumen width corresponding to capillaries (<20 µm) were manually counted on the whole slide for each slide (Figure 1A–C and Figure 6D). For each slide, 10 capillaries were selected, and regions of interest (ROIs) of their outer and inner surfaces were manually delineated (Figure 6E,F). For each structure, the surface area of the capillary wall was calculated (area of outer circumference − area of inner circumference), with average capillary wall thickness of 3.89 ± 0.056 μm standard error of the mean (SEM), and the mean fluorescence intensity value of the AQP1 or AQP4 antibody inside the capillary wall was analyzed (Figure 6G,H). To make the analysis more precise, a custom-made ImageJ macro was deployed on the area of the capillary wall, using “Max Entropy” thresholding criteria incorporated in ImageJ/Fiji software, https://imagej.net/plugins/maximum-entropy-threshold, accessed on 30 April 2025, utilizing the ImageJ/Fiji incorporated binary functions “Invert” and “Close-“and then employing the Analyze particles plugin incorporated in ImageJ/Fiji software, https://imagej.net/imaging/particle-analysis, accessed on 30 April 2025, using the following parameters: Size (micron’2): 0.2-Infinity, Circularity: 0.00-1.00, which gave us an ROI for each AQP cluster. The total number of clusters in each capillary was measured, along with the mean fluorescence intensity value of those AQP clusters and the percentage of the capillary wall populated with the clusters (Figure 6G,H).

To validate antibody specificity and confirm the absence of non-specific binding, positive and negative controls were performed for both AQP1 and CD31 immunolabeling. Kidney tissue served as a positive control, demonstrating characteristic AQP1 and CD31 expressions in glomerular and peritubular capillaries (Figure 6I). Liver tissue was used as a negative control, confirming the absence of specific immunoreactivity and validating antibody specificity (Figure 6J).

The study initially aimed to capture 10 AQP1-labeled capillaries per imaging field. However, due to the comparatively sparse capillary network in spinal cord sections, this target was adjusted, resulting in 527 cross-sectioned capillaries labeled with AQP1 antibodies and 505 labeled with AQP4 antibodies analyzed across three dura mater regions (parietal, falx, and spinal), totaling 1032 out of the planned 1200 capillaries.

### 4.6. Statistical Analysis

Statistical tests and *n*-values are indicated in figure captions. Statistical comparisons were performed using a one-way ANOVA test with multiple post hoc comparisons using the Tukey test for the correction of statistical hypothesis testing. All statistical tests were performed using GraphPad Prism 9 software. If *p* values were less than 0.05, the null hypothesis was rejected. Statistical values were expressed as means ± SEMs unless otherwise stated. * *p* < 0.05, ** *p* < 0.01, *** *p* < 0.001.

## 5. Conclusions

This study bridges structural anatomy and molecular physiology by mapping aquaporin expression in porcine dural capillaries. The coexistence of AQP1/AQP4 clusters with adjacent lymphatic networks provides a mechanistic framework for the Bulat–Klarica–Orešković hypothesis, emphasizing the dura’s active role in neurofluid homeostasis. However, it does not, by itself, indicate the directionality of neurofluid exchange across the capillary wall. These insights open new avenues for understanding meningeal contributions to CNS health and disease.

## Figures and Tables

**Figure 1 ijms-26-07653-f001:**
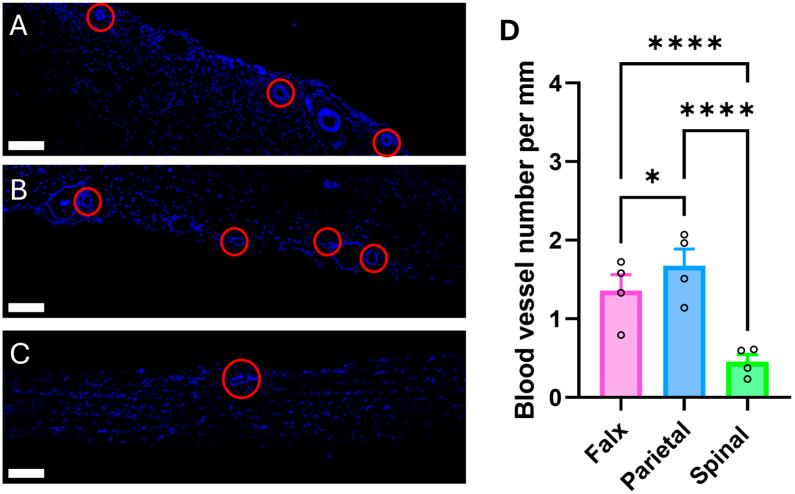
Distribution and quantification of CD31-positive blood vessels in distinct regions of the dura mater. (**A**–**C**) Representative confocal images of the (**A**) falx cerebri, (**B**) parietal dura, and (**C**) spinal dura, immunolabeled for CD31 (blue). Capillaries (diameter < 20 µm) are indicated by red circles. All images acquired at 10× objective; scale bars, 500 µm. (**D**) Quantification of capillary density, expressed as the mean number of capillaries per mm of dura mater in each region (mean ± SEM). Capillary density is significantly higher in the parietal dura (1.67 ± 0.21 vessels/mm) compared to the falx cerebri (1.36 ± 0.20 vessels/mm) and spinal dura (0.45 ± 0.09 vessels/mm). Statistical significance determined by two-way ANOVA with post hoc testing (* *p* < 0.05, **** *p* < 0.0001).

**Figure 2 ijms-26-07653-f002:**
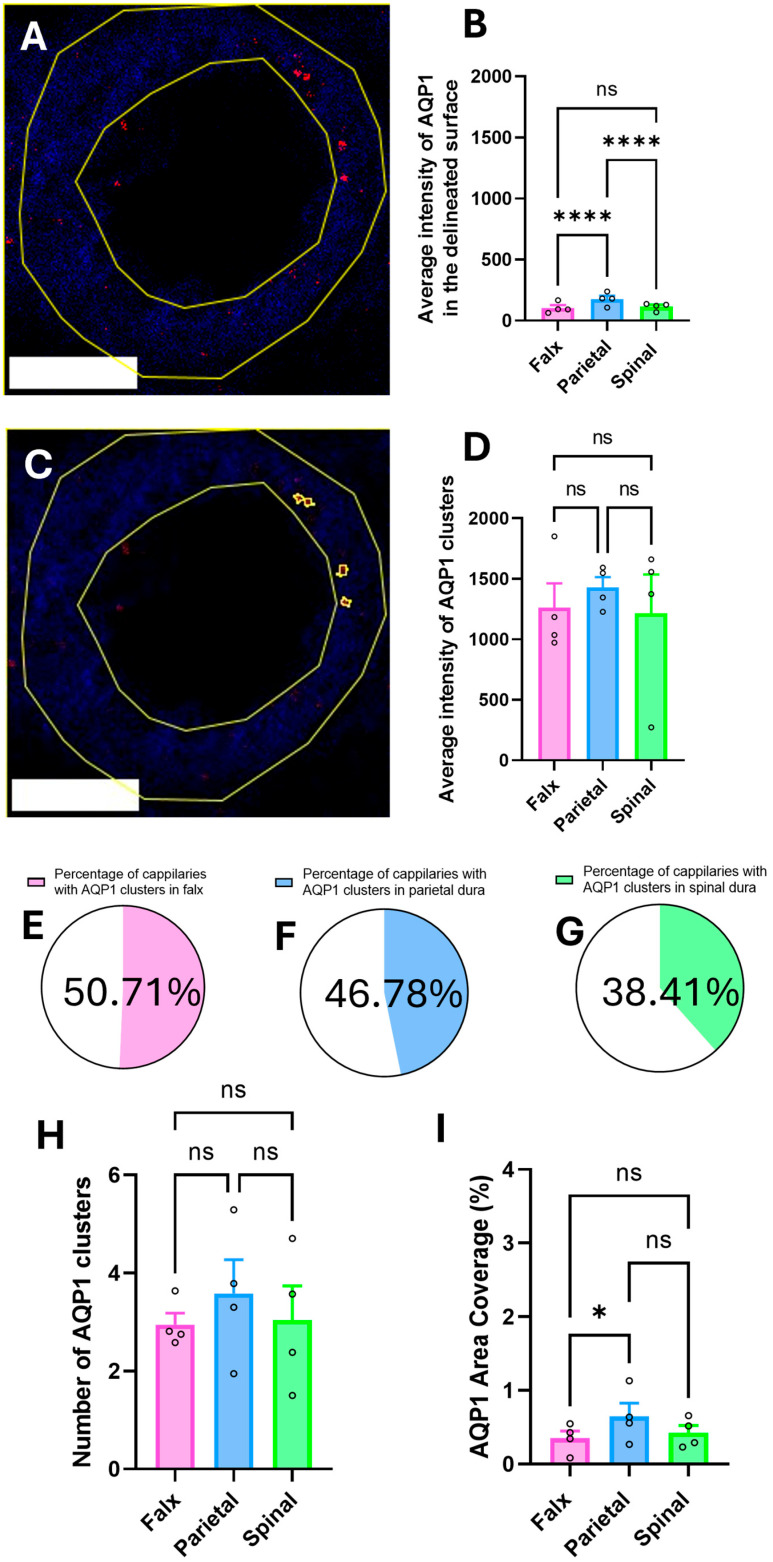
Regional distribution and intensity of AQP1 signal within delineated surface (capillary walls) of the dura mater. (**A**) Representative confocal image of a capillary immunolabeled for CD31 (blue), with the endothelial cytoplasmic surface area outlined by the outer and inner vessel boundaries delineated (capillary wall) (yellow). AQP1 immunoreactivity (red) is visible within the capillary wall; 20× objective, scale bar 20 μm. (**B**) Quantification of average intensity of AQP1 in the delineated surface across regions. AQP1 intensity is significantly higher in the parietal dura (177.4 ± 27.04) compared to the falx (104.7 ± 22.11) and spinal dura (114.9 ± 15.82), with no significant difference between the latter two regions (**** *p* < 0.0001, ns). (**C**) 20× objective image of the capillary shown in (**A**), highlighting AQP1-positive clusters (circled in yellow) within the capillary wall. Scale bar 20 μm. (**D**) Quantification of average AQP1 cluster intensity reveals no significant regional differences between the falx (1394 ± 205.3), parietal dura (1428 ± 85.78), and spinal dura (1549 ± 414.5) (ns). (**E**–**G**) Proportion of analyzed capillaries exhibiting AQP1 clusters in the falx (50.71%), parietal dura (46.78%), and spinal dura (38.41%), respectively, indicating regional variability in the presence of AQP1 clustering. (**H**) Average number of AQP1 clusters per capillary (where clusters were detected) does not differ significantly between regions: falx (2.95 ± 0.24), parietal dura (3.58 ± 0.69), and spinal dura (3.04 ± 0.70) (ns). (**I**) AQP1 area coverage (%) in each region. The parietal dura exhibits a significantly higher proportion of AQP1-positive wall area (0.65 ± 0.18%) compared to the falx (0.35 ± 0.10%, * *p* < 0.05), while the spinal dura (0.10 ± 0.2%) shows no significant difference relative to other regions. Statistical significance was determined by one-way ANOVA with Tukey’s post hoc test.

**Figure 3 ijms-26-07653-f003:**
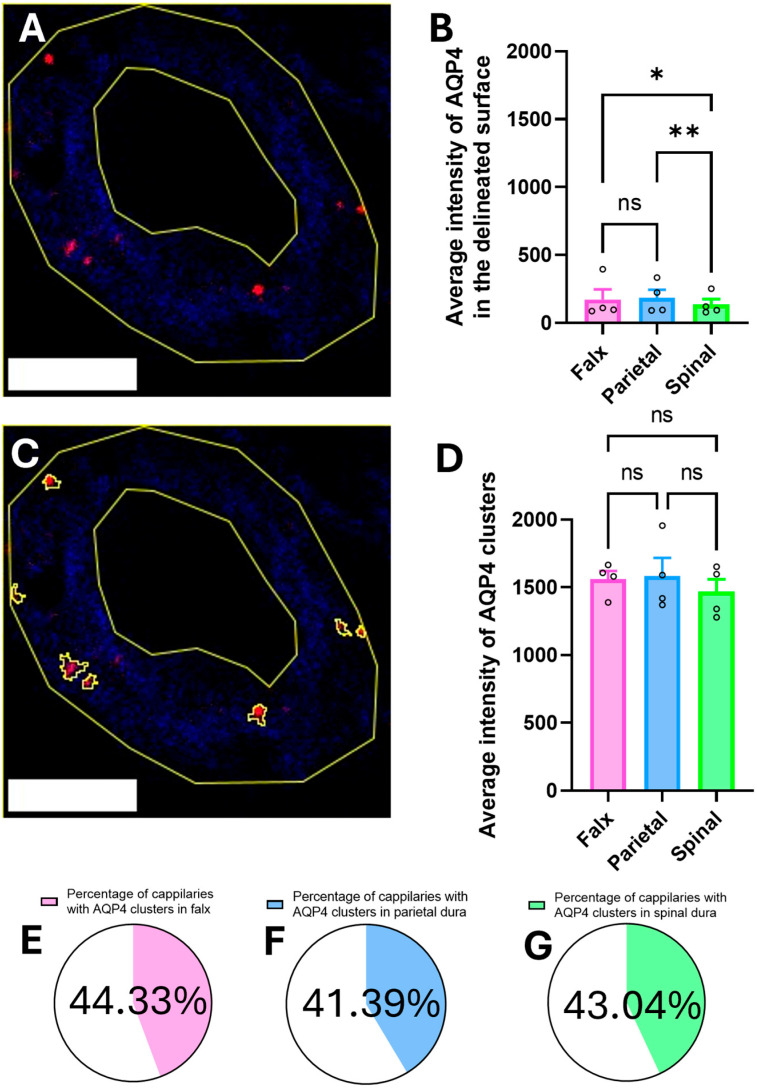
Regional distribution and intensity of AQP4 signal within delineated surface (capillary walls) of the dura mater. (**A**) Representative confocal image of a capillary immunolabeled for CD31 (blue), with the endothelial cytoplasmic surface area outlined by the outer and inner vessel boundaries delineated (capillary wall) (yellow). AQP4 immunoreactivity (red) is visible within the capillary wall. Scale bar 20 μm. (**B**) Quantification of average intensity of AQP4 in the delineated surface across regions. AQP4 intensity is significantly higher in the parietal dura (176.2 ± 46.01) compared to the spinal dura (135.8 ± 30.40, ** *p* < 0.01) and between falx (166.7 ± 57.87) and spinal dura (* *p* < 0.05), with no significant difference between falx and parietal regions (ns). (**C**) Image of the capillary shown in (**A**), highlighting AQP4-positive clusters (circled in yellow) within the capillary wall. (**D**) Quantification of average AQP4 cluster fluorescence intensity reveals no significant regional differences between the falx (1593 ± 56.19), parietal dura (1574 ± 103.1), and spinal dura (1453 ± 73.12) (ns). (**E**–**G**) Proportion of analyzed capillaries exhibiting AQP4 clusters in the falx (44.33%), parietal dura (41.39%), and spinal dura (43.04%), respectively. (**H**) Average number of AQP4 clusters per capillary (where clusters were detected) does not differ significantly between regions: falx (3.02 ± 0.86), parietal dura (3.35 ± 0.93), and spinal dura (2.74 ± 0.55) (ns). (**I**) AQP4 area coverage (%) in each region. The parietal dura exhibits a significantly higher proportion of AQP4-positive wall area (1.33 ± 0.57%) compared to the spinal dura (0.57 ± 0.20%, * *p* < 0.05), while no significant differences were observed between falx (0.83 ± 0.45%) and other regions. Data are presented as means ± SEM. Statistical significance was determined by one-way ANOVA with Tukey’s post hoc test.

**Figure 4 ijms-26-07653-f004:**
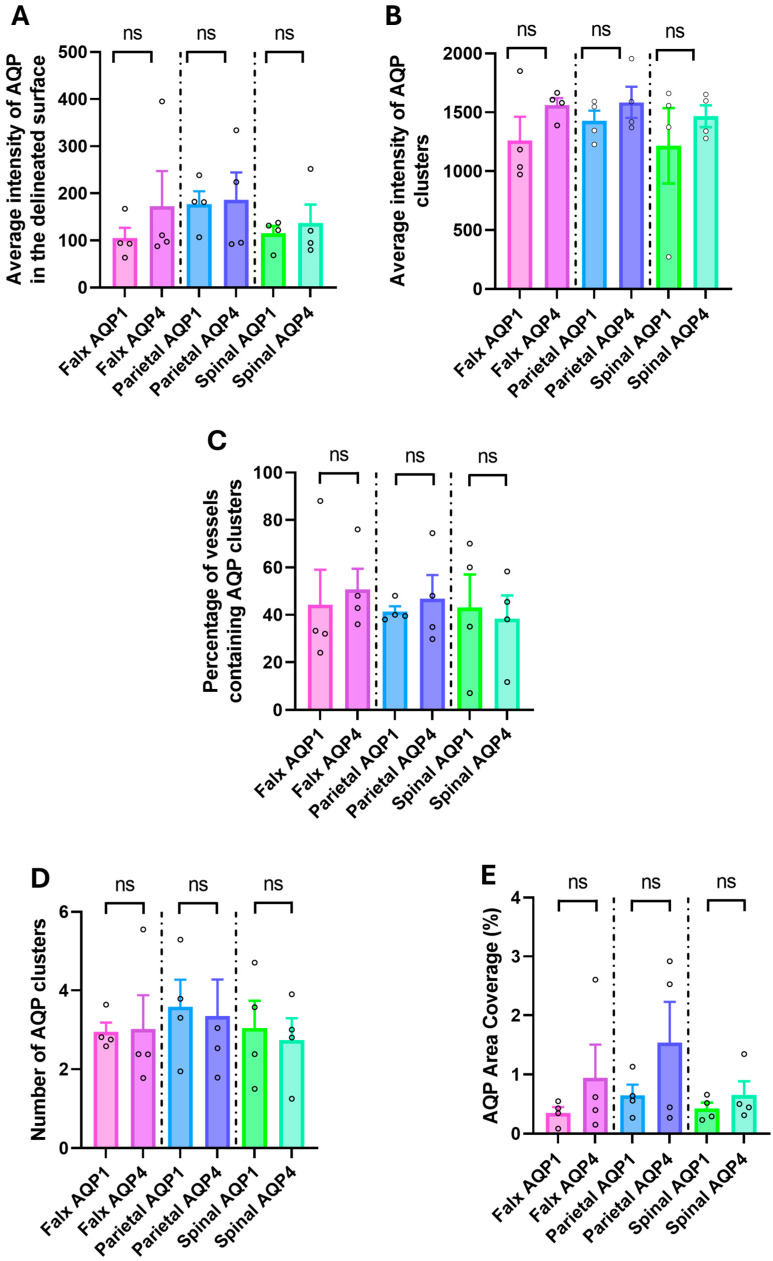
Comparative analysis of AQP1 and AQP4 distribution patterns across dural regions. Direct comparisons between AQP1 (pink/blue/light green bars) and AQP4 (purple/dark blue/green bars) within each anatomical region for multiple parameters. (**A**) Average intensity of AQP signal in the delineated capillary wall surface shows no significant differences between AQP1 and AQP4 in any region. (**B**) Average intensity of AQP clusters reveals no significant differences between the two aquaporins across all regions. (**C**) Percentage of vessels containing AQP clusters shows similar distribution patterns for both AQP1 and AQP4. (**D**) Number of AQP clusters per capillary demonstrates no significant differences between aquaporins in any region. (**E**) AQP area coverage (%) shows comparable proportions of capillary wall area occupied by AQP1 and AQP4 across all regions. Data presented as mean ± SEM. Statistical significance was determined by unpaired *t*-test for each regional comparison. ns = not significant (*p* > 0.05). Falx = falx cerebri; Parietal = parietal dura; Spinal = spinal dura.

**Figure 5 ijms-26-07653-f005:**
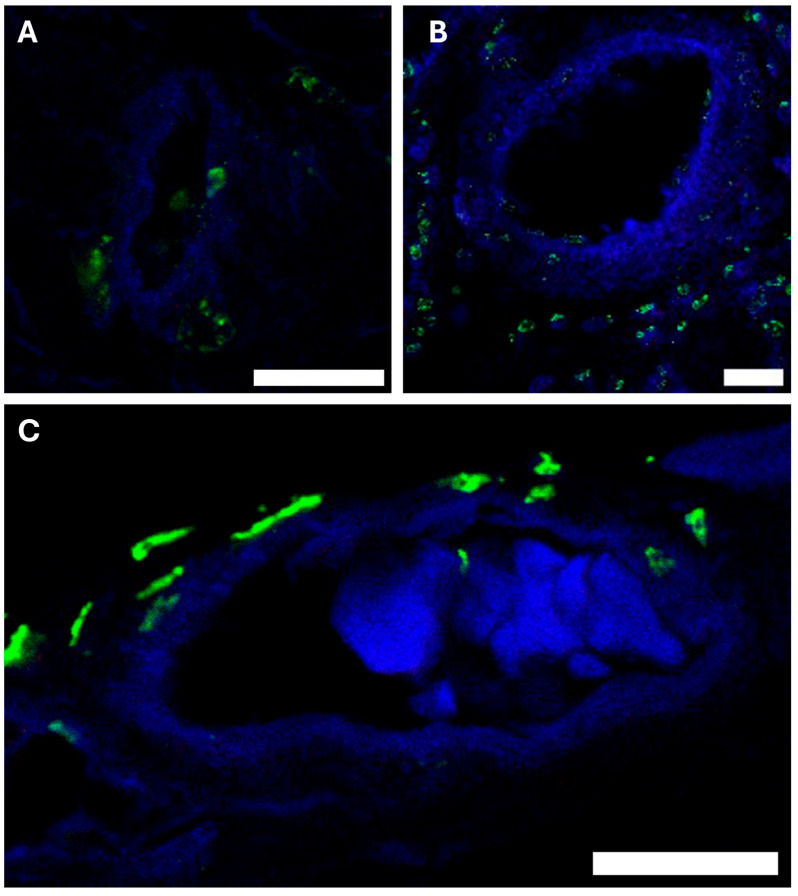
Immunofluorescence characterization of vascular and lymphatic structures in porcine dura mater. Confocal microscopy images showing podoplanin (PDPN, green) and CD31 (blue) double immunolabeling demonstrating the relationship between capillaries and lymphatic vessels across different dural regions. (**A**) Spinal dura mater imaged with 5× digital zoom. (**B**) Falx cerebri imaged without digital zoom, showing a larger vessel. (**C**) Parietal dura mater imaged with 5× digital zoom. The differential expression pattern confirms that the analyzed capillary structures are blood vessels (CD31-positive, PDPN-negative) rather than lymphatic vessels (PDPN-positive). Scale bar = 20 μm for all panels.

**Figure 6 ijms-26-07653-f006:**
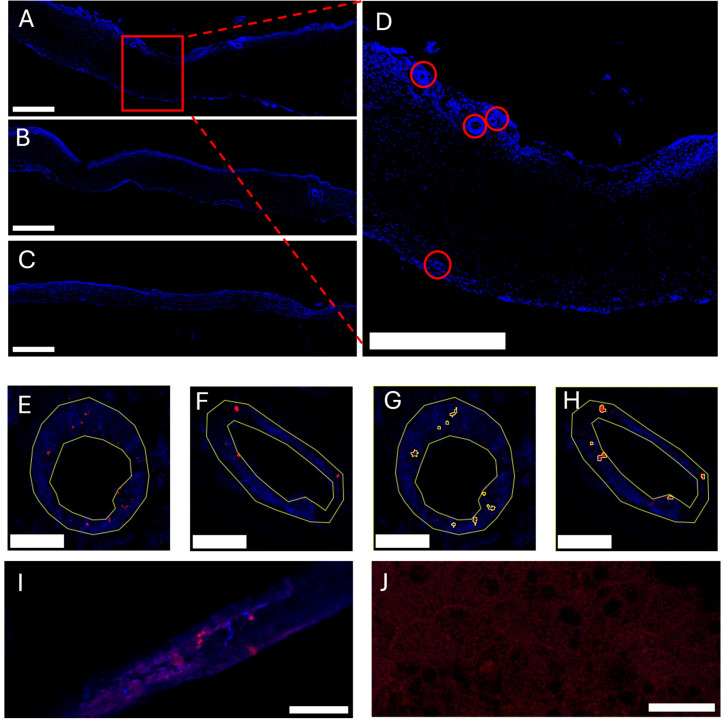
Confocal microscopy images of dura mater cross-sections demonstrating capillary architecture as noted by CD31 antibody reactivity (blue) and aquaporin expression (red). (**A**) Falx cerebri (cranial dura mater) at 10× objective. (**B**) Parietal dura mater. (**C**) Spinal dura mater. All images acquired at 10× objective; scale bars, 500 µm. (**D**) Higher magnification of the cranial dura (region indicated in (**A**)—red square), highlighting capillaries (circled in red). Scale bar = 100 µm. (**E**) Transverse section of a capillary with delineated outer and inner vessel walls (yellow), showing aquaporin 1 (AQP1) immunoreactivity within the vessel wall (red). Scale bar = 20 µm, objective 20×. (**F**) Transverse section of a capillary with delineated vessel walls (yellow), showing aquaporin 4 (AQP4) immunoreactivity (red). Scale bar = 20 µm. (**G**) Detail of the capillary shown in (**E**) with discrete AQP1-positive clusters annotated (yellow regions of interest (ROI-s) inside the delineated surface (capillary wall)). Scale bar = 20 µm. (**H**) Detail of the capillary shown in (**F**) with discrete AQP4-positive clusters annotated. Scale bar = 20 µm. (**I**) Positive control for AQP1 and CD31 immunoreactivity in kidney tissue, demonstrating specific AQP1 expression (red) and CD31 labeling (blue) in glomerular and peritubular capillaries. Scale bar = 20 µm. (**J**) Negative control for AQP1 and CD31 immunoreactivity in liver tissue, showing absence of specific signal, confirming antibody specificity. Scale bar = 20 µm.

## Data Availability

The data that support the findings of this study are available from the corresponding author upon reasonable request.

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
