# Peer review of "Aquaporins in the Capillaries of the Dura Mater of Pigs"

_ijms, 2025, doi:10.3390/ijms26157653_

Round 1

Reviewer 1 Report

Comments and Suggestions for Authors

The authors demonstrated convincingly the presence of capillary in porcine dura mater and the presence of aquaporins in the capillary wall. As to the conclusions drawn from the results, some additional thoughts are required:

  • Since aquaporins facilitate water flow in both directions, their presence alone doesn't allow any conclusion as to in which direction CSF is passing the capillary wall in dura mater
  • As the authors noted in the manuscript, the total number of capillary does matter. In addition, the total surface area of the capillary wall in relation to the surface area of the choroid plexus also matters regarding the relative contribution of these strutures to the total CSF production

Author Response

Comments 1: Since aquaporins facilitate water flow in both directions, their presence alone doesn't allow any conclusion as to in which direction CSF is passing the capillary wall in dura mater.

Response 1: Thank you for this important remark. We fully agree with the comment. Therefore, we have explicitly addressed this limitation in the Conclusion. We now clearly state that the identification of aquaporins indicates the capacity for water permeability but, by itself, does not reveal the directionality of cerebrospinal fluid (CSF) exchange across the capillary wall. This clarification is emphasized in the revised Conclusion (page 18, paragraph 5, line 541).

Comments 2: As the authors noted in the manuscript, the total number of capillary does matter. In addition, the total surface area of the capillary wall in relation to the surface area of the choroid plexus also matters regarding the relative contribution of these structures to the total CSF production.

Response 2: Thank you for pointing this out. We agree with this comment and have further emphasized this issue in the revised Discussion. While our primary focus was on aquaporin expression and regional capillary density, we now clearly note the importance of considering the total surface area of dural capillaries in comparison to the choroid plexus when assessing their relative contribution to CSF production. Importantly, we wish to clarify that our findings relate to the possible involvement of aquaporins in neurofluids transport processes. Our results do not challenge the role of the choroid plexus as an component of the CSF system, but rather suggest that other vascular surfaces, may also participate in neurofluid exchange alongside the choroid plexus.

Reviewer 2 Report

Comments and Suggestions for Authors

In the study Aquaporins in the capillaries of the dura mater of pigs by Martinovic et al., the authors present a comprehensive study on the expression pattern of Aqp4 and Aqp1 in several dura mater regions in the 3 months old pigs. They introduce and potentially reinforce the modern concept that connects CSF physiology with CNS interstitial fluid (ISF) and plasma in CNS microvessels fluid exchange.  

The study fills the critical gap in our knowledge of regional vascular characteristics within the dura mater providing the first comprehensive comparison of capillary density and Aqp expression between cranial and spinal regions in large animal model.  High-resolution confocal microscopy and quantitative immunofluorescence analyses across over 1,000 capillaries, revealed significantly higher capillary density and aquaporin expression in cranial dura. In addition, the identification of adjacent lymphatic structures supports a dual-system model of dural fluid regulation.

These findings offer important insights into CNS fluid dynamics and related disorders, such as subdural hematomas and hydrocephalus, and provide a strong foundation for future translational research, supporting the case for publication.

Minor comments:

The study was done on the 3-month-old pigs. It will benefit the reader to clarify which developmental stage this corresponds to – juvenile or early adult?  Is the described Aqp expression pattern dependent on the age of the animal?

 The labeling of the Fig.2A,B,C is missing in the text.

Author Response

Comments 1: The study was done on the 3-month-old pigs. It will benefit the reader to clarify which developmental stage this corresponds to – juvenile or early adult? Is the described Aqp expression pattern dependent on the age of the animal?

Response 1: Thank you for this valuable comment. We have clarified in the revised Methods and Discussion sections that the animals used were juvenile (3-month-old) pigs, corresponding to an early developmental stage before sexual maturity. We also recognize that age-dependent differences in aquaporin expression are possible and now explicitly identify this as a limitation and a direction for future studies. This clarification can be found in the revised Methods (page 3, paragraph 2.1, line 100) and Discussion (page 18, paragraph 4, line 530).

Comments 2: The labeling of the Fig.2A,B,C is missing in the text.

Response 2: Thank you for pointing out this oversight. We have carefully reviewed the manuscript and corrected all omitted or unclear figure references. The labeling for Fig. 2A, 2B, and 2C has now been included in the appropriate section of the Methods (page 7, paragraph 2.5, line 216).

Round 2

Reviewer 1 Report

Comments and Suggestions for Authors

Comments adequately addressed in the revised manuscript.